# Effects of Supplementation of Different Antioxidants to Cryopreservation Extender on the Post-Thaw Quality of Rooster Semen—A Meta-Analysis

**DOI:** 10.3390/ani14202936

**Published:** 2024-10-11

**Authors:** Esther Díaz Ruiz, Francisco Javier Navas González, José Manuel León Jurado, Ander Arando Arbulu, Juan Vicente Delgado Bermejo, Antonio González Ariza

**Affiliations:** 1Department of Genetics, Faculty of Veterinary Medicine, University of Cordoba, 14014 Cordoba, Spain; estherddrr@gmail.com (E.D.R.); fjng87@hotmail.com (F.J.N.G.); anderarando@hotmail.com (A.A.A.); juanviagr218@hotmail.com (J.V.D.B.); 2Institute of Agricultural Research and Training (IFAPA), Alameda del Obispo, 14004 Cordoba, Spain; 3Agropecuary Provincial Centre, Diputación Provincial de Córdoba, 14071 Cordoba, Spain; jmlj01@dipucordoba.es

**Keywords:** antioxidants, cryopreservation, rooster, semen

## Abstract

**Simple Summary:**

The standardization of the poultry semen cryopreservation technique is necessary for the conservation of genetic resources; however, during this process, a series of cellular lesions are generated as a consequence of the increase in reactive oxygen species (ROS). The objective of the present study is to determine which antioxidants are most commonly used in roosters to mitigate this effect after their addition to the cryopreservation diluent by studying different characteristics of seminal quality. A discriminant canonical analysis (DCA) was performed, and the variables hypo-osmotic swelling test (HOST), viability, and total motility were reported to be the ones that provided the most information, as they were the most discriminant variables between groups. According to the descriptive statistics, antioxidants catalase and serine were the ones that gave the best results in most of the variables studied. In conclusion, given that catalase had the most beneficial effects on the seminal quality parameters that proved to have the greatest discriminating power, this antioxidant could be the one of choice.

**Abstract:**

The standardization of the semen cryopreservation technique could be an effective tool in poultry for the conservation of genetic resources. During this process, the production of reactive oxygen species increases, leading to oxidative stress that causes damage to the spermatozoa. To reduce this effect, the addition of exogenous antioxidants in the cryopreservation diluent has been reported to be effective. Multiple antioxidants such as catalase, vitamin E, cysteamine, ergothioneine, and serine have been studied in roosters. Therefore, the present investigation aims to perform a meta-analysis to determine if the use of the aforementioned antioxidants added to the cryopreservation extender produces an improvement in semen quality parameters in roosters after thawing. After collecting the data, a discriminant canonical analysis was performed to determine which of the selected semen quality traits provided the most information, with hypo-osmotic swelling test (HOST), viability, and total motility variables showing the highest discriminatory power. However, according to the descriptive statistics, catalase and serine are the antioxidants that improve a greater number of seminal quality parameters, and since catalase gives the most favorable results for most of the discriminating variables, it could be the antioxidant of choice.

## 1. Introduction

Cryopreservation of avian sperm is a fundamental tool that allows the conservation of genetic resources of both endangered and more productive breeds, leading to an increase in biodiversity [1]. However, during the cryopreservation process, various types of cellular damage occur, and spermatozoa from many species, including avian species, are particularly sensitive to oxidative stress. This stress leads to decreased motility and DNA damage, which ultimately affects fertility [2].

Oxidative stress is generated when there is an imbalance between the production of reactive oxygen species (ROS) and their elimination by the endogenous antioxidant system present in the avian seminal plasma [3], and is considered one of the main factors leading to poor sperm quality [4]. To reduce oxidative stress, the addition of exogenous antioxidants in the cryopreservation extender could be beneficial by interrupting the oxidative chain reaction that is generated [5].

Several studies have evaluated the effect of supplementation of cryopreservation extender with antioxidants in roosters, including catalase [6,7,8], vitamin E [6,7,9,10], cysteamine [11,12,13], ergothioneine [11,14], and serine [11,15]. The vast majority of these articles have been carried out in Iran, whose interest in this area may be because poultry breeding in that country dates back to ancient Persia around 2000–2500 BC when chicken species were first introduced [16]. Moreover, the poultry industry nowadays plays a key role in Iran [17]. Thus, the development of assisted reproduction techniques is important to maintain genetic diversity. This diversity is mainly achieved through the development of germplasm banks [18]. This explains the interest in optimizing the cryopreservation process by using antioxidants in the cryopreservation extender.

The mechanism of action of catalase in cells involves neutralizing the toxicity caused by hydrogen peroxide (H_2_O_2_) by converting it into oxygen (O_2_) and water (H_2_O)” [19]. In addition to the roosters, this antioxidant has been tested in other species such as pigs [20], cattle [21], cats [22], sheep and goats [23], among others, being the main effects caused an improvement of sperm motility and viability, being also able to increase the potential of in vitro fertilization in the case of pigs.

Lipid-soluble vitamin E is recognized as a principal component of the endogenous antioxidant system in spermatozoa. It plays a crucial role in preventing lipid peroxidation of the sperm membrane by neutralizing various free radicals generated by reactive oxygen species (ROS), including superoxide anions, hydroxyl radicals, and hydrogen peroxide. This action contributes to improved sperm quality following thawing [10,24,25]. This antioxidant has been widely used in other animal species, such as the cat, in which good semen quality parameters have been obtained by improving sperm motility and post-thaw plasma membrane integrity [26].

In the case of cysteamine, it is an aminothiol antioxidant capable of eliminating the hydroxyl radical, improving glutathione synthesis, and contributing to the maintenance of the redox state in the oocytes [27,28,29]. This antioxidant has also been applied in other animal species, improving the quality of cryopreserved sperm in sheep and goats, resulting in an increase in sperm motility and a decrease in the number of abnormal spermatozoa [30,31].

Ergothioneine is a hydrophilic amino acid derived from histidine with a high antioxidant property capable of scavenging the most active free radicals [32]. Its main function is to protect spermatozoa from oxidative stress that occurs during the cryopreservation process by counteracting the negative effects that peroxides generate on sperm viability [33]. This substance has been evaluated in species other than roosters, such as rams, in which an improvement in sperm motility and sperm DNA integrity parameters after thawing was observed [34,35]. In dogs and pigs, its addition to the cryopreservation extender has also been proven to be beneficial in improving motility and kinematic parameters [36,37].

Finally, serine is a conditionally non-essential amino acid found in the seminal plasma of roosters, its presence in high amounts being favorable since it is directly related to the preservation of DNA integrity in thawed semen [38]. The antioxidant capacity of this compound could be because it favors the synthesis of glutathione and the methionine cycle, which contributes by reducing oxidative stress [39]. Unlike the other antioxidants contemplated in the present study, there are no bibliography references that report its use in other species different from roosters. This antioxidant has been reported to improve motility and kinematic parameters, membrane and acrosome integrity, and mitochondrial membrane potential [11].

In this sense, the objective of the present work is to perform a meta-analysis on the effect of the supplementation of different antioxidants to the cryopreservation extender on the post-thaw quality of rooster semen in terms of kinematic parameters, sperm viability, and membrane functionality, as well as to determine the correlation between these semen quality variables.

## 2. Materials and Methods

### 2.1. Research Strategy

In the present work, the systematic review strategy used has been previously described and reported to be an efficient tool for the meta-analysis of databases in the animal science field [40,41,42].

The data collection period covers approximately 12 years, from October 2010 to April 2022, inclusive. This period was chosen because October 2010 was the month and year in which the oldest research document was published, and April 2022 was the time when the research sampling process was completed. Data collection was carried out in Google Scholar (https://scholar.google.com; accessed on 31 July 2024) and Scopus (https://www.scopus.com; accessed on 31 July 2024) databases without any filter in terms of language or year of publication, selecting all the articles that studied the use of antioxidants in the cryopreservation extender of rooster semen. The keywords used were antioxidant, rooster, semen, seminal quality, and cryopreservation. The selection of papers was carried out based on their relevance to the use of antioxidants in diluents for cryopreservation of rooster semen, and no further quality filters were applied because factors such as those related to indexing were recorded as variables within the original database [43].

To obtain a more solid and robust database, those variables with more than 150 observations were used in the present study: total motility (TM, %), progressive motility (PM, %), curvilinear velocity (VCL, µm/s), straight-line velocity (VSL, µm/s), average path velocity (VAP, μm/s), linearity (LIN, %), straightness (STR, %), viability (SYBR-14/PI assay) and membrane functionality (Hypo-osmotic swelling test; HOST). As for the choice of antioxidants, those with the highest number of observations (more than 10) in the original database were used. The antioxidants that remained in the database were catalase, cysteamine, ergothioneine, serine, vitamin E, and those in which no antioxidants were used (control group).

Given that in the evaluated bibliography, there are different units of measurement for the variables contemplated in the present study, all of them were standardized and converted to the most frequent units in the studied papers. As for the methodologies used in each study to obtain the different semen quality parameters, they have not been recorded since these methods have been previously standardized to comply with adequate research procedures. Although there may be differences between the different techniques, these would be insignificant [40].

### 2.2. Data Analysis

#### 2.2.1. Discriminant Canonical Analysis (DCA)

As aforementioned, for DCA, 9 explanatory variables were included: TM, PM, VCL, VSL, VAP, LIN, STR, viability, and HOST. The different antioxidants studied (catalase, cysteamine, ergothioneine, serine, vitamin E, and no antioxidants) were used as classification criteria.

Forward stepwise multinomial logistic regression algorithms were used to perform the canonical variable selection. Priors were regularized according to group sizes computed using the a priori probability of SPSS Statistics for Windows software (Version 26.0, 2017, IBM Corp., Armonk, NY, USA) rather than considering them equal to avoid groups with different sample sizes affecting the quality of the classification [44].

According to several authors [45,46], to mitigate possible distortion effects, the minimum required sample size is at least 20 observations for each of 4 or 5 predictors, with the maximum number of independent variables being *n*-2, so, according to this premise, an adequate sample size has been used in the present analysis.

The discriminant canonical analysis routine of the analyzing data package of XLSTAT software, version 2022.4.1 (Addinsoft Pearson Edition 2022, Addinsoft, Paris, France) was used to perform the discriminant canonical analysis.

#### 2.2.2. Multicollinearity Preliminary Testing

Multicollinearity analyses were performed before the statistical analyses to rule out linear relationships between the different predictors and to ensure the independence of the variables contemplated in the study. This can detect redundancy problems between variables before data manipulation to ensure that the variance of the explanatory potential is not over-inflated by redundancy problems [40]. The variance inflation factor (VIF) is an indicator of multicollinearity whose values above 5 are not recommended [47] and is calculated by using the following formula:(1)$$VIF=1/(1-R^{2})$$
where *R*^2^ represents the coefficient of determination of the regression equation and tolerance (1 − *R*^2^) reflects the degree of variability in a specific independent variable that is not explained by the rest, whose recommended values are under 0.20 [48]. To perform the multicollinearity test, the multicollinearity statistics routine of the describing data package of XLSTAT software (Addinsoft Pearson Edition 2022, Addinsoft, Paris, France) was used.

#### 2.2.3. DCA Efficiency and Analysis Model Reliability

Pillai’s trace criterion is the only acceptable test to evaluate the assumption of equal covariance matrices in cases of unequal sample sizes [49]. A subroutine of the discriminant analysis routine of the analyzing data package of XLSTAT software (Addinsoft Pearson Edition 2022, Addinsoft, Paris, France) was used to calculate this parameter, and the set of predictors of the DCA was considered statistically significant when the significance was ≤0.05 [45]. The evidence that the set of predictors has a statistically significant effect on the values of the response variable is greater as the value of Pillai’s trace increases, and this statistic is considered the most robust for general protection against deviations from normality and homogeneity of variance of the multivariate residuals.

#### 2.2.4. Independent Factor Discriminant Potential Evaluation

Since the residuals (observed values) have to follow a normal distribution, the one-dimensional test for equality of class means is performed to test the difference in means between the different antioxidants after the elimination of redundant variables. The higher the F value, and therefore the lower the Wilks’ Lambda values, the better the discriminatory power of the variables. It was computed as a subroutine of the discriminant analysis routine of the analyzing data package of the XLSTAT 2022 software (Addinsoft Pearson Edition 2022, Addinsoft, Paris, France).

#### 2.2.5. Correlation Matrix

A correlation matrix between the different semen quality traits was created and graphically represented through a color map elaborated through the web server Heatmapper (www.heatmapper.ca; accessed on 4 June 2024).

#### 2.2.6. Canonical Coefficients and Loading Interpretation and Spatial Representation

Using DCA, the percentage assignment of a sample within its group (defined by the different antioxidants studied) was computed. Discriminant loading values of ≥|0.40| for a variable can be considered significantly discriminant. Thus, when the discriminant power of that variable is higher, the value of the absolute coefficient increases.

#### 2.2.7. Discriminant Function Cross-Validation

The leave-one-out cross-validation was used as this method can validate the discriminant functions. When the classification index is at least 25% higher than that obtained by chance, the classification is considered to be accurate [45]. The Press’ Q significance test was employed to compare the discriminating power using the following formula:Press’ Q = [n − (n′K)]^2^/[n (K − 1)],(2)
where n is the number of observations in the sample, n′ is the number of observations correctly classified, and K is the number of groups.

The value of the Press’ Q statistic should be compared with the critical value of 6.63 for χ^2^ with a degree of freedom in a significance level of 0.01, with the cross-validated classification being considered significantly better than chance when the Press’ Q value exceeds the critical value of χ^2^ = 6.63.

#### 2.2.8. Overall Descriptive Statistics

The mean of each antioxidant evaluated (catalase, cysteamine, ergothioneine, serine, vitamin E, and no antioxidant) was established for the different selected semen quality traits that remained in the DCA such as TM, PM, VCL, VSL, LIN, STR, viability and HOST. To perform this analysis, we used the descriptive statistics routine of the data description package of XLSTAT software (Addinsoft Pearson Edition 2022, Addinsoft, Paris, France).

## 3. Results

### 3.1. Multicollinearity Preliminary Test

A single round was required for the set of explanatory variables considered in the present study to reach an acceptable VIF level (VIF ≤ 5). After performing the multicollinearity analysis, the variable VAP with a VIF value of 19.198 was eliminated, considered redundant, and discarded for subsequent analyses. The variables that remained in the analysis whose VIF value was <5 are shown in Table 1.

### 3.2. DCA Efficiency and Analysis Model Reliability

Significant Pillai’s trace criterion determined the validity of the discriminant canonical analysis (Table 2).

Significant discriminant capacities were reported for two of the five functions revealed after discriminant analysis, with the discriminatory power of functions F1 and F2 being high (eigenvalue of 1.528 and 0.370, respectively), with 93.971% of the variance being explained by these two functions (Figure 1).

### 3.3. Independent Factor Discriminant Potential Evaluation

The discriminant capacity of the different variables contemplated in the analysis is determined by the value of F and Wilks’ Lambda, with their discriminant power being higher as the value of F increases and Wilks’ Lambda decreases. This allowed us to perform a ranking of variables according to their discriminant power. The present analysis revealed that HOST, viability, TM, LIN, VCL, STR, and VSL contributed significantly (*p* < 0.05) to the discriminant functions (Table 3).

### 3.4. Correlation Matrix

The results of the correlation matrix between the semen quality parameters are shown in Figure 2, which shows values between +0.771 and −0.289.

### 3.5. Canonical Coefficients and Loading Interpretation and Spatial Representation

Figure 3 reports discriminant canonical coefficient loadings for representative variables across discriminant functions. HOST (coefficient value = |0.890|), viability (coefficient value = |0.725|), and TM (coefficient value = |0.568|) were the traits with the highest standardized canonical discriminant function coefficients for F1. VCL (coefficient value = |0.707|), LIN (coefficient value = |0.629|), and TM (coefficient value = |0.539|) were the traits that showed the highest coefficient values for F2.

Figure 4 shows a clear differentiation between the antioxidants under study. To determine the relative position of the centroids, it is necessary to substitute the mean value of the observations in each term of the first two discriminant functions (F1 and F2) to obtain the coordinates of the x and y axes. The greater the distance between the centroids, the greater the predictive power of the canonical discriminant function to classify the observations.

### 3.6. Discriminant Function Cross-Validation

A Press’ Q value of 942.72 (n = 205; n′ = 198; K = 6) was computed. Thus, predictions can be considered to be better than chance at 95%. Table 4 shows the cross-validation of discriminant classification results. The discriminant tool developed in the present study allowed us to correctly classify 96.59% of the studied samples.

### 3.7. Overall Descriptive Statistics

The mean for each semen quality parameter studied according to the antioxidant used is shown in Table 5. Catalase and serine were the antioxidants that offered the most desirable quality results, with catalase showing higher results for TM, viability, and HOST and serine for PM, VCL, VSL, LIN, and STR.

## 4. Discussion

The use of antioxidants in the poultry species should be studied in depth since these species are highly susceptible to the excessive production of ROS that occurs during the cryopreservation process due to the large amount of fatty acids present in the membrane, especially docosatetraenoic acid (22:4n − 6) and arachidonic acid (20:4n − 6), which causes membrane peroxidation [25,50].

In the present study, the effect of various antioxidants on post-thaw semen quality in roosters was analyzed by analyzing several semen quality parameters. To develop an effective predictive model, we analyzed the relationships between various explanatory variables, selecting those that uniquely contribute to explaining the variability of the data. Variables exhibiting multicollinearity, indicated by a VIF greater than 5, were excluded from further analysis. In this sense, the only variable that reported multicollinearity problems was VAP. This variable could be redundant with other kinematic parameters, such as VSL and VCL, which are also velocity parameters. However, VSL and VCL measure velocity over specific trajectories, being VSL considered the most accurate parameter for determining sperm velocity in roosters [51]. In addition, a strong correlation between VAP and TM has been reported in this species [52].

After performing the DCA, the variables that significantly contributed to determining differences between groups were TM, HOST, viability, LIN, VCL, STR, and VSL. However, according to their position in the ranking (Table 3), the variables with the greatest discriminating power were HOST, viability, and TM.

HOST was the variable with the highest discriminant power. This fact has been previously reported in other studies in which the effect of antioxidants in the cryopreservation extender on the post-thaw semen quality of rooster semen was studied [53,54]. The plasma membrane of spermatozoa acts as a protective barrier during the cryopreservation process. So, an adequate structural membrane integrity that maintains its dynamic properties is essential for the sperm to function properly and achieve optimal fertility results. In addition, this structure is a key element in other physiological processes of the sperm that determine the success of fertilization, such as motility, capacitation, acrosomal reaction, and fusion with the oocyte [55,56]. The fact that the avian species has a large number of PUFAs compared to other mammalian species makes the HOST test a very important tool for determining semen quality in roosters [57,58]. Thus, spermatozoa subjected to a hypoosmotic environment show different swelling patterns ranging from minimal coiling of the tail tip to complete coiling encompassing the midpiece [59].

Another variable with great discriminating power is sperm viability. This trait must have adequate values for the poultry industry to progress reproductively [60]. In addition, its measurement is considered especially important in the presence of low sperm motility since a distinction must be made between spermatozoa that are alive but not motile and those that are dead [61]. Sperm viability is a variable that is severely affected by sperm storage and freezing. Thus, a balance must be achieved between ROS production and the scavenging capacity of the antioxidant system, either endogenously or through the application of exogenous substances [62]. For measuring sperm viability, the method of choice would be through the SYBR-14/PI assay since staining time is not as critical as when esterase-based stains are used and because background staining is virtually nonexistent [63].

Finally, TM is also another variable with high discriminatory power. Rooster sperm need to have adequate quality in terms of motility to reach the sperm storage tubules (SSTs) present in the hen’s reproductive tract [62]. In this regard, the determination of post-thaw sperm motility plays a pivotal role since cells that have been damaged during the cryopreservation process will result in reduced sperm fertility, as sperm motility is directly related to fertilization potential [64]. To preserve this function properly, mitochondrial activity must be preserved and is necessary for the correct maintenance of sperm function and energy metabolism through ATP synthase and oxidative phosphorylation [65]. As reported in a previous study on the rooster species, those samples with higher mitochondrial activity gave rise to higher motility results [66].

In any case, the control of the rest of the seminal quality parameters is necessary since they will determine the reproductive capacity of the males and allow for carrying out adequate artificial insemination programs [67]. For this purpose, the correlation matrix is a useful tool that allows the study of the existing correlations between the different seminal quality parameters contemplated in our study.

A high positive correlation is described between VCL/STR (+0.771), viability/HOST (+0.743), TM/viability (+0.711), and VSL/LIN (+0.706).

The positive correlation between VCL and STR may be because they are reliable parameters for determining the fertilizing potential of spermatozoa before their use in artificial insemination. According to Amann and Waberski [68], the relationship that exists between both parameters allows classifying spermatozoa as non-motile when VCL is close to 0, undesirable movement when VCL or STR presents a low value, or satisfactory movement when VCL and STR present adequate values.

VSL and TM variables showed a negative correlation. Spermatozoa must have adequate movement to reach the fertilization site. This motility is not only defined by the speed but also by the quality of the movement, which has to show progressive mobility with a minimum value of speed [69]. A spermatozoon that moves in circles and presents a large curvature in its trajectory but moves quickly or a spermatozoon that presents a linear trajectory but its movement is slow is unlikely to be competitive [70]. In a test performed on chickens by Froman [71], the mobility of spermatozoa was measured according to their ability to penetrate an Accudenz solution. A VSL value higher than 30 µm/s was reported to be needed to penetrate this solution. Therefore, an ejaculate needs minimum values of VSL to reach the SSTs in the female.

Finally, previous authors have reported that despite the effect of the cryopreservation process on sperm motility in roosters, the integrity of the sperm membrane has not been compromised [72], which would explain the negative correlation between VSL and sperm viability that has been previously reported in this species [52].

In short, this correlation study between the different variables would allow for improving the semen quality evaluation protocol since the exclusion of variables that showed a positive correlation would reduce the cost and time invested in analyzing redundant parameters, which would lead to designing a more efficient evaluation protocol.

By paying attention to the descriptive statistics (Table 5), the supplementation of the cryopreservation extender in roosters with catalase improved the results of TM, viability, and HOST. This coincides with a study performed in sheep semen in which TM and viability were increased after the addition of catalase in the cryopreservation diluent, but no such effects were observed when supplemented with cysteamine [73]. Likewise, in a study carried out in dogs, catalase was reported to have a more evident effect on the improvement of sperm quality by increasing the values of sperm motility, viability, and functionality [74]. This makes sense since catalase is an important enzyme system possessed by sperm to stop ROS damage by degrading hydrogen peroxide (H_2_O_2_), a free radical generated during the cryopreservation process, into oxygen and water [75].

On the other hand, the addition of serine causes the most favorable values for PM, VCL, VSL, LIN, and STR. Serine could improve kinematic parameters due to changes in sperm motility that are directly related to the phosphorylation of flagellar proteins such as serine [76]. Phosphorylation is fundamental in many cellular aspects, with protein kinases being important regulators of cellular processes such as movement. These proteins possess a catalytic domain that transfers a phosphate group from ATP and covalently binds it to specific amino acids containing a free hydroxyl group in their composition, such as serine [77].

By contrast, in the present study, cysteamine and vitamin E were the antioxidants that gave rise to the worst seminal quality results, with low TM, PM, and HOST results for cysteamine and low VSL, LIN, and STR results for vitamin E. These results are in agreement with another study conducted on buffalos since the supplementation of cryopreservation extender with cysteamine negatively affected motility between other parameters, such as acrosome morphology and integrity [78]. The same results were reported on bull species. Cysteamine supplementation resulted in a significant reduction of TM, PM, and acrosome integrity in thawed semen [79].

Finally, supplementation with vitamin E analogs on semen quality has also been evaluated. The inclusion of Trolox has not shown positive effects on sperm motility, viability, or morphology. When BHT has been added, a reduction of these same parameters has been reported [80]. This is partly in agreement with the results obtained in the present work since vitamin E has resulted in low results on kinematic parameters, but satisfactory results have been observed for TM, viability, and HOST. Moreover, supplementation of the extender with vitamin E has been reported to have favorable results in humans [81], cattle [82], and sheep [83]. However, for motility parameters, vitamin E has reported discordant results [26,84,85].

## 5. Conclusions

In conclusion, the addition of antioxidants in the cryopreservation extender has been reported to be a useful tool to improve the quality of thawed rooster semen and allows the development of improved assisted poultry reproduction techniques. Of the five antioxidants more used in avian sperm cryopreservation, catalase and serine were the two that gave rise to the best results in post-thawed semen quality in roosters. Although serine improves motility and kinematic parameters generally, catalase is the preferred antioxidant. Catalase yields more favorable results in terms of HOST, viability, and TM—seminal quality parameters that demonstrated better discriminating power in evaluating the effectiveness of various antioxidants in rooster semen.

## Figures and Tables

**Figure 1 animals-14-02936-f001:**
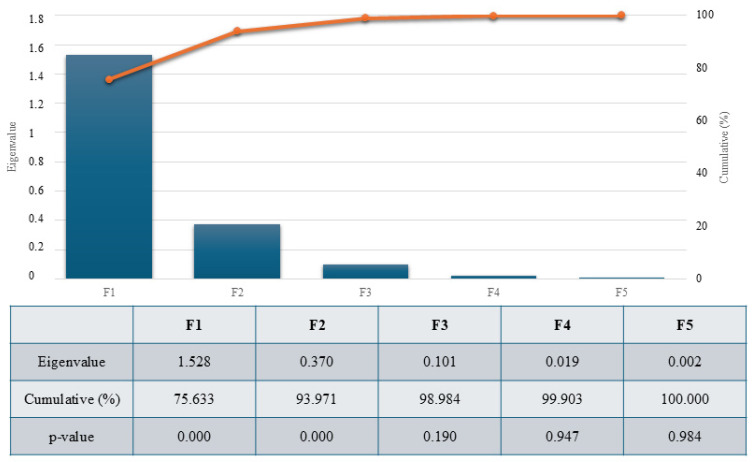
Canonical variable functions and percentages of self-explained and cumulative variance. The eigenvalue of each discriminant function is used to measure each function’s discriminative power.

**Figure 2 animals-14-02936-f002:**
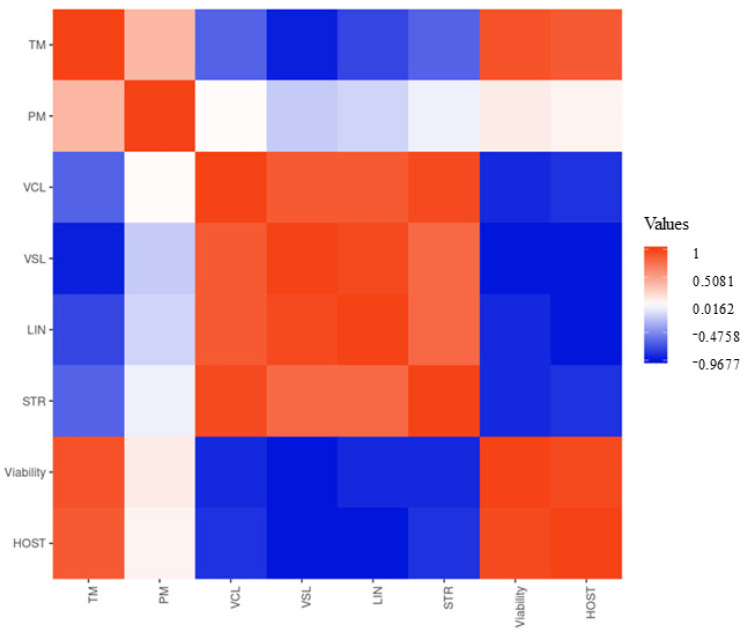
Correlation matrix between the different semen quality parameters studied. TM: total motility; PM: progressive motility; VCL: curvilinear velocity; VSL: straight-line velocity; LIN: linearity; STR: straightness; HOST: hypo-osmotic swelling test.

**Figure 3 animals-14-02936-f003:**
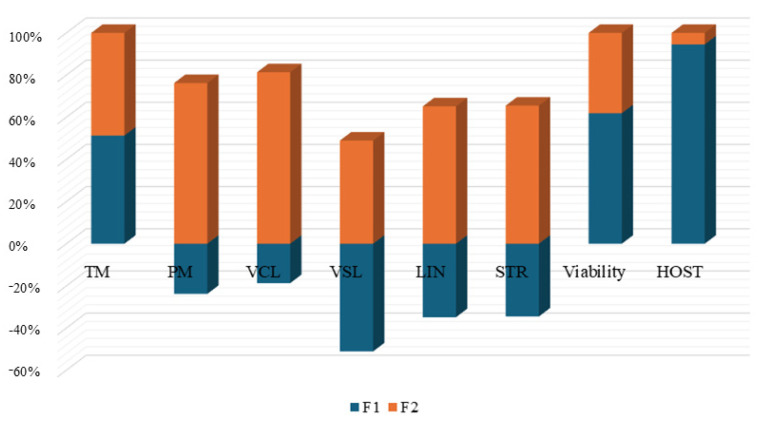
Standardized canonical discriminant function coefficients. TM: total motility; PM: progressive motility; VCL: curvilinear velocity; VSL: straight-line velocity; LIN: linearity; STR: straightness; HOST: hypo-osmotic swelling test.

**Figure 4 animals-14-02936-f004:**
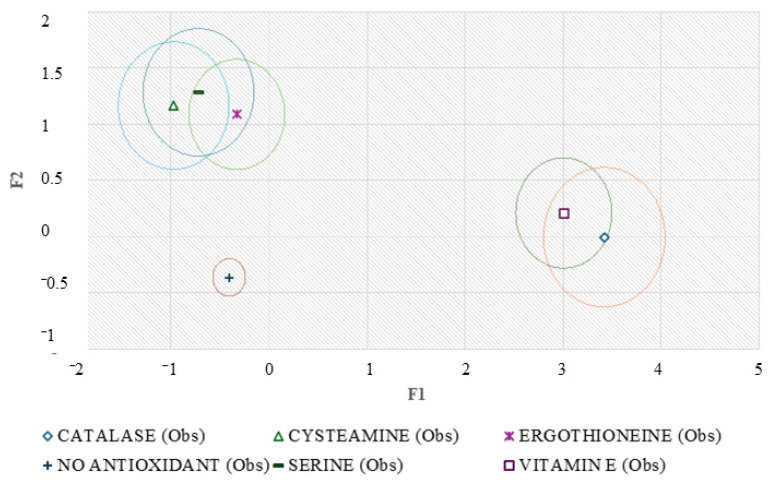
Territorial map depicting the results of the canonical discriminant analysis on the different antioxidants studied. The larger the differences between each point, the better the predictive power of the DCA classifying them.

**Table 1 animals-14-02936-t001:** Multicollinearity analysis of sperm quality-related traits.

	Tolerance (1 − R^2^)	VIF
**VCL (µm/s)**	0.212	4.727
**TM (%)**	0.240	4.162
**VSL (µm/s)**	0.323	3.095
**Viability (%)**	0.330	3.030
**LIN (%)**	0.355	2.814
**HOST (%)**	0.357	2.801
**STR (%)**	0.383	2.609
**PM (%)**	0.570	1.755

Interpretation rule of thumb: variance inflation factor (VIF) = 1 (not correlated); 1 < VIF < 5 (moderately correlated); VIF ≥ 5 (highly correlated). VIFs > 5 are not presented in the table. VCL: curvilinear velocity; TM: total motility; VSL: straight-line velocity; LIN: linearity; HOST: hypo-osmotic swelling test; STR: straightness; PM: progressive motility.

**Table 2 animals-14-02936-t002:** Summary of the results of Pillai’s trace of equality of covariance matrices of canonical discriminant functions.

Trace	F (Observed Value)	F (Critical Value)	DF1	DF2	*p*-Value	Alpha
0.987	6.025	1.407	40	980	<0.0001	0.05

F, Snedecor’s F; df1, numerator degrees of freedom for the F approximation; df2, denominator degrees of freedom for the F approximation.

**Table 3 animals-14-02936-t003:** Results for the unidimensional test of equality of the means of the classes to test for difference in the means across sample groups once redundant variables have been removed.

Variable	Rank	Wilks’ Lambda	F	DF1	DF2	*p*-Value
HOST **(%)**	1	0.514	37.624	5	199	<0.0001
Viability **(%)**	2	0.622	24.176	5	199	<0.0001
TM **(%)**	3	0.719	15.535	5	199	<0.0001
LIN **(%)**	4	0.805	9.648	5	199	<0.0001
VCL **(µm/s)**	5	0.821	8.653	5	199	<0.0001
STR **(%)**	6	0.850	7.020	5	199	<0.0001
VSL **(µm/s)**	7	0.901	4.374	5	199	0.001
PM **(%)**	8	0.962	1.572	5	199	0.170

F, Snedecor’s F; df1, numerator degrees of freedom for the F approximation (groups minus 1); df2, denominator degrees of freedom for the F approximation (observations minus 1). HOST: hypo-osmotic swelling test; TM: total motility; LIN: linearity; VCL: curvilinear velocity; STR: straightness; VSL: straight-line velocity; PM: progressive motility.

**Table 4 animals-14-02936-t004:** Cross-validation of classification results.

From\To	Catalase	Cysteamine	Ergothioneine	No Antioxidant	Serine	Vitamin E	Total	% Correct
**Catalase**	10	0	0	0	0	0	10	100.00
**Cysteamine**	0	12	0	0	0	0	12	100.00
**Ergothioneine**	0	2	14	0	0	0	16	87.50
**No antioxidant**	0	1	0	134	0	4	139	96.40
**Serine**	0	0	0	0	12	0	12	100.00
**Vitamin E**	0	0	0	0	0	16	16	100.00
**Total**	10	15	14	134	12	20	205	96.59

**Table 5 animals-14-02936-t005:** Means by class.

	Catalase	Cysteamine	Ergothioneine	Serine	Vitamin E	No Antioxidant
**TM (%)**	72.641	45.957	54.404	59.625	72.158	40.295
**PM (%)**	15.149	14.762	21.813	25.658	15.209	16.485
**VCL (µm/s)**	55.952	77.302	85.253	109.351	56.919	59.416
**VSL (µm/s)**	18.207	31.735	35.856	46.902	17.531	29.493
**LIN (%)**	31.872	45.783	48.983	56.241	31.196	38.445
**STR (%)**	56.380	70.117	72.396	90.666	58.216	64.801
**Viability (%)**	76.641	46.288	53.651	43.077	74.726	38.226
**HOST (%)**	79.910	27.625	40.295	39.093	73.288	37.352

TM: total motility; PM: progressive motility; VCL: curvilinear velocity; VSL: straight-line velocity; LIN: linearity; STR: straightness; HOST: hypo-osmotic swelling test.

## Data Availability

All data stemming from the present research are enclosed in the tables and figures. Any additional data will be made accessible from the corresponding authors upon reasonable request.

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
