# Peer review of "Effects of Supplementation of Different Antioxidants to Cryopreservation Extender on the Post-Thaw Quality of Rooster Semen—A Meta-Analysis"

_animals, 2024, doi:10.3390/ani14202936_

Round 1

Reviewer 1 Report

Comments and Suggestions for Authors

This review paper recapitulates the previous findings of research studies based on the effects of different antioxidants on the post-thaw (PT) quality of rooster semen. The Reviewer suggests that the following comments need to be addressed before the paper can be considered for publication.

Comments

1.Title

Should consider to rephrase it. Possible suggestion: ²Effects of supplementation of different antioxidants to cryopreservation extender on the post-thaw quality of rooster semen - a meta……² The specific role of each antioxidant on the different sperm attributes following freezing-thawing was not analyzed in this study.

2. Objectives (L101-104). The objectives need to be more clearly defined (in accordance with the title). Which traits (L104)?

3. M & M

a) A major concern of this study was the strategy used to select the publications (L111-122). If the data were obtained in April 2022, how long was the collection period, 1 year, 2 years, etc? Why a filtering system was not used? Besides the original papers, were review papers, case reports, etc included?

Results

a) Re-check the  relevance of the phylogenetic tree clustering shown in Figure 5  (p. 8)!.

b) Tables and Figures. Most of captions/legends of the tables and  figures are poorly described. The captions need to provide more information about the tables/figures, for example, give the full names of the abbreviations. The data shown in Table 5 are unreadable (L390-391).

 Discussion

a) The three most important sperm attributes that seemed to be of significant relevance in this study are HOST, viability and total motility. The Authors should be aware that HOST is a poor predictor of PT semen quality because it is only limited to the functional integrity of the membrane overlying the tail region of the sperm (L306-316?), whereas sperm viability was not fully defined in this study (L317-324?). Sperm viability in different research studies could be referring to the mitochondrial function (JC-1/PI assay), or the structural integrity of the plasma membrane (SYBR-14/PI assay is commonly used for this analysis).

b) Why the results of fertility outcome following cryopreservation of antioxidant-supplemented rooster semen were not included in this study (L313/L351)?

c) Should consider to remove these statements because they are not relevant to the Discussion (L348-371).

d) Give the source of hydrogen peroxide (L398), and expand more about the role of serine.

e) Should consider to remove the last statement -L432-434.

Comments on the Quality of English Language

Some minor corrections are required.

Reviewer 2 Report

Comments and Suggestions for Authors

Dear Authors,

Congratulations on the completed study. Given the extensive range of antioxidants available for improving semen quality post-thawing, such a meta-analysis is essential and provides significant assistance in identifying the best options. The article is well written and leads to precise and highly relevant conclusions. The primary limitation is that the statistical methods employed are somewhat complex for the general readership, which might diminish interest in the article. A few specific comments are provided below:

Line 19 don't use abbreviations before including them in text

Line 45 delete “and in the case of avian species, spermatozoa are particularly sensitive”, sensitivity to oxidative stress during cryopreservation is not unique to avian species. Something like this sound better “However, during the cryopreservation process, various cellular damages occur, and spermatozoa from many species, including avian species, are particularly sensitive to oxidative stress. This stress leads to decreased motility and DNA damage, which ultimately affects fertility.”  Lines 44-47

Lines 60-63 a reference is needed to support these statements

Line 64 replace “is the elimination of” with “involves neutralizing” and “these radicals” with “it”. Replace with this all sentence “The mechanism of action of catalase in cells involves neutralizing the toxicity caused by hydrogen peroxide (H2O2) by converting it into oxygen (O2) and water (H2O).”

Lines 66-67 Add sheep and goats to the enumeration. I suggest consulting the bibliographical source titled Advancements in Understanding and Enhancing Antioxidant-Mediated Sperm Cryopreservation in Small Ruminants: Challenges and Perspectives. This source provides a comprehensive discussion on the subject and could be considered for citation.

Lines 70-74 too long sentence, replace with this for clarity “Lipid-soluble vitamin E is recognized as a principal component of the endogenous antioxidant system in spermatozoa. It plays a crucial role in preventing lipid peroxidation of the sperm membrane by neutralizing various free radicals generated by reactive oxygen species (ROS), including superoxide anions, hydroxyl radicals, and hydrogen peroxide. This action contributes to improved sperm quality following thawing.”

Lines 291-295 replace with this for clarity “To develop an effective predictive model, we analyzed the relationships between various explanatory variables, selecting those that uniquely contribute to explaining the variability of the data. Variables exhibiting multicollinearity, indicated by a VIF greater than 5, were excluded from further analysis.”

Line 316 replace test with tool

Lines 427-432 replace with this for clarity “Although serine improves motility and kinematic parameters generally, catalase is the preferred antioxidant. Catalase yields more favorable results in terms of HOST, viability, and TM—seminal quality parameters that demonstrated better discriminating power in evaluating the effectiveness of various antioxidants in rooster semen.”

Comments on the Quality of English Language

Minor editing of English language required.

Reviewer 3 Report

Comments and Suggestions for Authors

This is an interesting study using various statistical models to define HOST, viability, and total motility as the most determinant variables in rooster semen quality after cryopreservation and thawing. Besides, among the antioxidants used as the extender in rooster semen cryopreservation, serine and particularly catalase were concluded to have the most discriminant powers giving to better semen quality.

1.      The simple summary is suggested to rewrite to briefly describe aims, major findings, and conclusions instead of looking like introduction.

2.      Among the various datasets used for studies, how the authors normalized the values of variables (such as HOST, viability, and total motility) that are assessed at different time points, different dilutions of sperms, various extenders, and storage for various durations. Besides, how the authors determined the dose effects, such as 300 μg/mL catalase vs. 0.05 mg/ml vitamin E.

Comments on the Quality of English Language

Minor editing of English language required.

Round 2

Reviewer 1 Report

Comments and Suggestions for Authors

The Authors have addressed all my comments, except for a few points.

1. Change to ²post-thaw quality², where necessary throughout the text (L105).

2. Approx. 12 years of the data collection, that is from October 2010 to April 2022 (L114). Re-check.

3. There are still some inconsistencies with the sperm quality parameter. In L129, it indicates that the HOS test represents sperm viability and membrane functionality. It is misleading because sperm viability is presented as an individual assay in the Results, tables, etc. Give the assay used to assess the sperm viability if it was performed individually (L129).

Reviewer 3 Report

Comments and Suggestions for Authors

The concerns have been answered.

Comments on the Quality of English Language

Minor editing of English language required

Author Response

Dear reviewer, 

The manuscript has undergone extensive revision of English and grammar.

To this end, we thank you for your time and attention and for considering this manuscript.ç

Yours faithfully,
Dr. Antonio González Ariza